# Relational Graph Attention Networks

## Abstract

We investigate Relational Graph Attention Networks, a class of models that extends non-relational graph attention mechanisms to incorporate relational information, opening up these methods to a wider variety of problems. A thorough evaluation of these models is performed, and comparisons are made against established benchmarks. To provide a meaningful comparison, we retrain Relational Graph Convolutional Networks, the spectral counterpart of Relational Graph Attention Networks, and evaluate them under the same conditions. We find that Relational Graph Attention Networks perform worse than anticipated, although some configurations are marginally beneficial for modelling molecular properties. We provide insights as to why this may be, and suggest both modifications to evaluation strategies, as well as directions to investigate for future work.

## 1 Introduction

Convolutional Neural Networks (CNNs) successfully solve a variety of tasks in Euclidean grid-like domains, such as image captioning (Donahue et al., 2017) and classifying videos (Karpathy et al., 2014). CNNs are successful because they assume the data is locally stationary and compositional (Defferrard et al., 2016; Henaff et al., 2015; Bruna et al., 2013).

However, data often occurs in the form of graphs or manifolds, which are classic examples of non-Euclidean domains. Specific instances include knowledge bases, molecules, and point clouds captured by 3D data acquisition devices (Wang et al., 2018). The generalisation of Neural Networks (NNs) to non-Euclidean domains is termed Geometric Deep Learning (GDL), and may be roughly divided into spectral, spatial and hybrid approaches (Bronstein et al., 2017).

Spectral approaches (Defferrard et al., 2016), most notably Graph Convolutional Networks (GCNs) (Kipf and Welling, 2016), are limited by their basis-dependence. A filter that is learned with respect to a basis on one domain is not guaranteed to behave similarly when applied to another basis and domain. Spatial approaches are limited by an absence of shift invariance and lack of coordinate system (Duvenaud et al., 2015; Atwood and Towsley, 2016; Monti et al., 2017). Hybrid approaches combine spectral and spatial approaches, trading their advantages and deficiencies against each-other (Bronstein et al., 2017; Rustamov and Guibas, 2013; Szlam et al., 2005; Gavish et al., 2010).

A recent approach that began with Graph Attention Networks (GATs), applied attention mechanisms to graphs, and does not share these limitations (Veličković et al., 2017; Gong and Cheng, 2018; Zhang et al., 2018; Monti et al., 2018; Lee et al., 2018).

An alternative direction has been to generalise Recurrent Neural Networks (RNNs) from sequential message passing on one-dimensional signals, to message passing on graphs (Sperduti and Starita, 1997; Frasconi et al., 1997; Gori et al., 2005). Incorporating gating mechanisms led to the development of Gated Graph Neural Networks (GGNNs) (Scarselli et al., 2009; Allamanis et al., 2017)[1].

Relational Graph Convolutional Networks (RGCNs) have been proposed as an extension of GCNs to the domain of relational graphs (Schlichtkrull et al., 2018). This model has

---

[1]We note that GGNNs support relation types. Evaluating these models on the tasks presented here is necessary to acquire a better understanding neural models of relational data.

achieved impressive performance on node classification and link prediction tasks, however, its mechanisms still resides within spectral methods and shares their deficiencies. The focus of this work investigate generalisations of RGCN away from its spectral origins.

We take RGCN as a starting point, and investigate a class of models we term Relational Graph Attention Networks (RGATs), extending attention mechanisms to the relational graph domain. We consider two variants, Within-Relation Graph Attention (WIRGAT) and Across-Relation Graph Attention (ARGAT), each with either additive or multiplicative attention. We perform an extensive hyperparameter search, and evaluate these models on challenging transductive node classification and inductive graph classification tasks. These models are compared against established benchmarks, as well as a re-tuned RGCN model.

We show that RGAT performs worse than expected, although some configurations produce marginal benefits on inductive graph classification tasks. In order to aid further investigation in this direction, we present the full Cumulative Distribution Functions (CDFs) for the hyperparameter searches in Appendix D, and statistical hypothesis tests in Appendix E. We also provide a vectorised, sparse, batched implementation of RGAT and RGCN in `TensorFlow` which is compatible with `eager` execution mode to open up research into these models to a wider audience[2].

## 2 RGAT ARCHITECTURE

### 2.1 RELATIONAL GRAPH ATTENTION LAYER

We follow the construction of the GAT layer in Veličković et al. (2017), extending to the relational setting, using ideas from Schlichtkrull et al. (2018).

**Layer input and output**   The input to the layer is a graph with $R = |\mathcal{R}|$ relation types and $N$ nodes. The $i^{\text{th}}$ node is represented by a feature vector $\boldsymbol{h}_i \in \mathbb{R}^F$, and the features of all nodes are summarised in the feature matrix $\boldsymbol{H} = [\boldsymbol{h}_1 \, \boldsymbol{h}_2 \, \ldots \, \boldsymbol{h}_N] \in \mathbb{R}^{N \times F}$. The output of the layer is the transformed feature matrix $\boldsymbol{H}' = [\boldsymbol{h}'_1 \, \boldsymbol{h}'_2 \, \ldots \, \boldsymbol{h}'_N] \in \mathbb{R}^{N \times F'}$, where $\boldsymbol{h}'_i \in \mathbb{R}^{F'}$ is the transformed feature vector of the $i^{\text{th}}$ node.

**Intermediate representations**   Different relations convey distinct pieces of information. The update rule of Schlichtkrull et al. (2018) made this manifest by assigning each node $i$ a distinct intermediate representation $\boldsymbol{g}_i^{(r)} \in \mathbb{R}^{F'}$ under relation $r$

$$\boldsymbol{G}^{(r)} = \boldsymbol{H} \, \boldsymbol{W}^{(r)} \in \mathbb{R}^{N \times F'}, \tag{1}$$

where $\boldsymbol{G}^{(r)} = \left[ \boldsymbol{g}_1^{(r)} \, \boldsymbol{g}_2^{(r)} \, \ldots \, \boldsymbol{g}_N^{(r)} \right]$ is the intermediate representation feature matrix under relation $r$, and $\boldsymbol{W}^{(r)} \in \mathbb{R}^{F \times F'}$ are the learnable parameters of a shared linear transformation.

**Logits**   Following Veličković et al. (2017); Zhang et al. (2018), we assume the attention coefficient between two nodes is based only on the features of those nodes up to a neighborhood-level normalisation. To keep computational complexity linear in $R$, we assume that, given linear transformations $\boldsymbol{W}^{(r)}$, the logits $E_{i,j}^{(r)}$ of each relation $r$ are independent

$$E_{i,j}^{(r)} = a \left( \boldsymbol{g}_i^{(r)}, \boldsymbol{g}_j^{(r)} \right), \tag{2}$$

and indicate the importance of node $j$'s intermediate representation to that of node $i$ under relation $r$. The attention is masked so that, for node $i$, coefficients $\alpha_{i,j}^{(r)}$ exist only for $j \in \mathcal{n}_i^{(r)}$, where $\mathcal{n}_i^{(r)}$ denotes the set of neighbor indices of node $i$ under relation $r \in \mathcal{R}$.

**Queries, keys and values**   The logits are composed from queries and keys, and specify how the values, i.e. the intermediate representations $\boldsymbol{g}_i^{(r)}$, will combine to produce the updated

---

[2]https://github.com/anonymous/rgat.

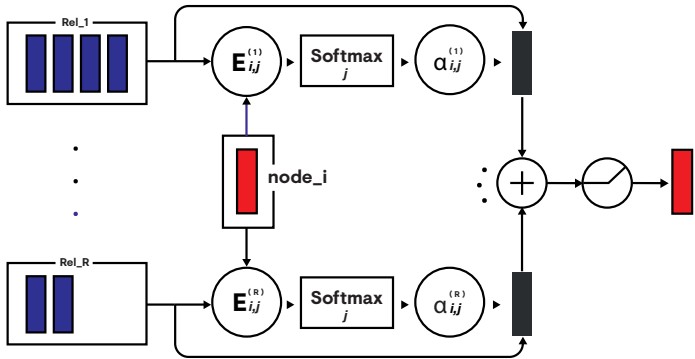

Figure 1: WIRGAT. The intermediate representations for node $i$ (left red rectangle) are combined with the intermediate representations for nodes in its neighborhood (blue rectangles) under each relation $r$, to form each logit $E_{i,j}^{(r)}$. A softmax is taken over each logit matrix for each relation type to form the attention coefficients $\alpha_{i,j}^{(r)}$. These attention coefficients construct a weighted sum over the nodes in the neighborhood for each relation (black rectangle). These are then aggregated and passed through a nonlinearity to produce the updated representation for node $i$ (right red rectangle).

node representations $\boldsymbol{h}_i'$ (Vaswani et al., 2017). A separate query kernel $\boldsymbol{Q}^{(r)} \in \mathbb{R}^{F' \times D}$ and key kernel $\boldsymbol{K}^{(r)} \in \mathbb{R}^{F' \times D}$ project the intermediate representations $\boldsymbol{g}_i^{(r)}$, into query and key representations of dimensionality $D$

$$\boldsymbol{q}_i^{(r)} = \boldsymbol{g}_i^{(r)} \boldsymbol{Q}^{(r)} \in \mathbb{R}^D, \qquad\qquad \boldsymbol{k}_i^{(r)} = \boldsymbol{g}_i^{(r)} \boldsymbol{K}^{(r)} \in \mathbb{R}^D. \qquad (3)$$

For convenience, the query and key kernels are combined to form the attention kernels $\boldsymbol{A}^{(r)} = \boldsymbol{Q}^{(r)} \oplus \boldsymbol{K}^{(r)} \in \mathbb{R}^{2F' \times D}$. These query and key representations are the building blocks of the two specific realisations of $a$ in Equation (2) that we now consider.

**Additive attention logits**    The first realisation of $a$ we consider is the relational modification of the logit mechanism of Veličković et al. (2017)

$$E_{i,j}^{(r)} = \text{LeakyReLu}\left(q_i^{(r)} + k_j^{(r)}\right), \qquad (4)$$

where the query and key dimensionality are both $D = 1$, and $q_i^{(r)}$ and $k_i^{(r)}$ are scalar flattenings of their one-dimensional vector counterparts $\boldsymbol{q}_i^{(r)}, \boldsymbol{k}_i^{(r)} \in \mathbb{R}^1$. We refer to any instance of RGAT using logits of the form in Equation (4) as additive RGAT.

**Multiplicative attention logits**    The second realisation we consider is the multiplicative mechanism of Vaswani et al. (2017); Zhang et al. (2018)[3]

$$E_{i,j}^{(r)} = \boldsymbol{q}_i^{(r)} \cdot \boldsymbol{k}_j^{(r)}, \qquad (5)$$

where the query and key dimensionality $D$ can be any positive integer. We refer to any instance of RGAT using logits of the form in Equation (4) as multiplicative RGAT.

It should be noted that there are many types of attention mechanisms beyond vanilla additive and multiplicative. These include mechanisms leveraging the structure of the dual graph Monti et al. (2018) as well as learned edge features Gong and Cheng (2018).

The attention coefficients should be comparable across nodes. This can be achieved by applying softmax appropriately to any logits $E_{i,j}^{(r)}$. We investigate two candidates, each encoding a different prior belief about how the importance of different relations.

---

[3]The form of our mechanism is not precisely that of Zhang et al. (2018) as they also consider residual concatenation and gating mechanism applied across the heads of the attention mechanism.

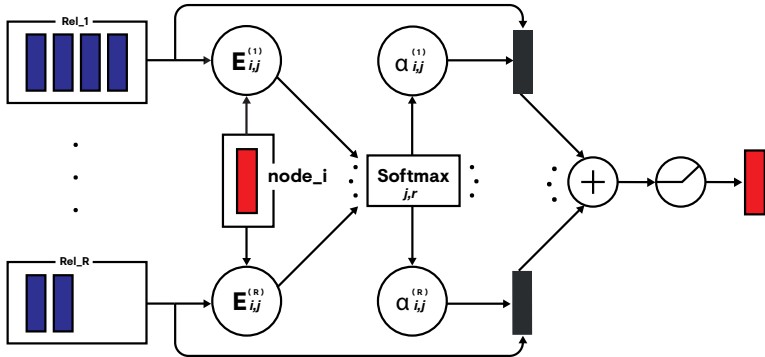

Figure 2: ARGAT. The logits are produced identically to those in Figure 1. A softmax is taken across all logits independent of relation type to form the attention coefficients $\alpha_{i,j}^{(r)}$. The remaining weighting and aggregation steps are the same as those in Figure 1.

**WIRGAT** The simplest way to take the softmax over the logits $E_{i,j}^{(r)}$ of Equation (4) or Equation (5) is to do so independently for each relation $r$

$$\alpha_{i,j}^{(r)} = \operatorname*{softmax}_{j}\left(E_{i,j}^{(r)}\right) = \frac{\exp\left(E_{i,j}^{(r)}\right)}{\sum_{k \in n_i^{(r)}} \exp\left(E_{i,k}^{(r)}\right)}, \qquad \forall\, i, r : \sum_{j \in n_i^{(r)}} \alpha_{i,j}^{(r)} = 1. \qquad (6)$$

We call the attention in Equation (6) Within-Relation Graph Attention (WIRGAT), and it is shown in Figure 1. This mechanism encodes the prior that relation importance is a purely global property of the graph by implementing an independent probability distribution over nodes in the neighborhood of $i$ for each relation $r$. Explicitly, for any node $i$ and relation $r$, nodes $j, k \in n_i^{(r)}$ yield competing attention coefficients $\alpha_{i,j}^{(r)}$ and $\alpha_{i,k}^{(r)}$ with sizes depending on their corresponding representations $\boldsymbol{g}_j^{(r)}$ and $\boldsymbol{g}_k^{(r)}$. There is no competition between any attention coefficients $\alpha_{i,j}^{(r)}$ and $\alpha_{i,k}^{(r')}$ for all nodes $i$ and nodes $j \in n_i^{(r)}, j' \in n^{(r')}$ where $r' \neq r$ irrespective of node representations.

**ARGAT** An alternative way to take the softmax over the logits $E_{i,j}^{(r)}$ of Equation (4) or Equation (5) is across node neighborhoods irrespective of relation $r$

$$\alpha_{i,j}^{(r)} = \operatorname*{softmax}_{j,r}\left(E_{i,j}^{(r)}\right) = \frac{\exp\left(E_{i,j}^{(r)}\right)}{\sum_{r' \in \mathscr{R}} \sum_{k \in n_i^{(r')}} \exp\left(E_{i,k}^{(r')}\right)}, \qquad \forall\, i : \sum_{r \in \mathscr{R}} \sum_{j \in n_i^{(r)}} \alpha_{i,j}^{(r)} = 1. \qquad (7)$$

We call the attention in Equation (7) Equation (6) Across-Relation Graph Attention (ARGAT), and it is shown in Figure 2. This mechanism encodes the prior that relation importance is a local property of the graph by implementing a single probability distribution over the different representations $\boldsymbol{g}_j^{(r)}$ for nodes j in the neighborhood of node $i$. Explicitly, for any node $i$ and all $r, r' \in \mathscr{R}$, all nodes $j \in n_i^{(r)}$ and $k \in n_i^{(r')}$ yield competing attention coefficients $\alpha_{i,j}^{(r)}$ and $\alpha_{i,k}^{(r')}$ with sizes depending on their corresponding representations $\boldsymbol{g}_j^{(r)}$ and $\boldsymbol{g}_k^{(r')}$.

**Comparison to RGCN** For comparison, the coefficients of RGCN are given by $\alpha_{i,j}^{(r)} = |n_i^{(r)}|^{-1}$. This encodes the prior that the intermediate representations of nodes $j \in n_i^{(r)}$ to node $i$ under relation $r$ are equally important.

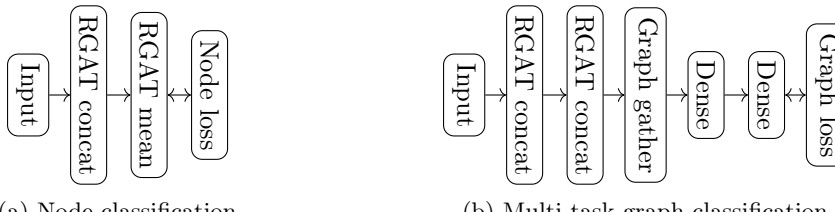

(a) Node classification.  (b) Multi-task graph classification.

Figure 3: (a) The network architecture used for node classification on AIFB and MUTAG. This architecture is the same as in Schlichtkrull et al. (2018) except with RGCNs replaced with RGATs. (b) The network architecture used for multi-task graph classification on Tox21. This architecture is the same as the GCNs architecture in Altae-Tran et al. (2016) except with RGCNs replaces with GATs and we do not use graph pooling.

**Propagation rule**  Combining the attention mechanism of either Equation (6) or Equation (7) with the neighborhood aggregation step of Schlichtkrull et al. (2018) gives

$$\boldsymbol{h}_i' = \sigma \left( \sum_{r \in \mathcal{R}} \sum_{j \in n_i^{(r)}} \alpha_{i,j}^{(r)} \, \boldsymbol{g}_j^{(r)} \right) \in \mathbb{R}^{N \times F'}, \tag{8}$$

where $\sigma$ represents an optional nonlinearity. Similar to Vaswani et al. (2017); Veličković et al. (2017), we also find that using multiple heads in the attention mechanism can enhance performance

$$\boldsymbol{h}_i' = \bigoplus_{k=1}^{K} \sigma \left( \sum_{r \in \mathcal{R}} \sum_{j \in n_i^{(r)}} \alpha_{i,j}^{(r,k)} \, \boldsymbol{g}_j^{(r,k)} \right) \in \mathbb{R}^{N \times K F'}, \tag{9}$$

where $\oplus$ denotes vector concatenation, $\alpha_{i,j}^{(r,k)}$ are the normalised attention coefficients under relation $r$ computed by either WIRGAT or ARGAT, and $\boldsymbol{g}_i^{(r,k)} = \boldsymbol{h}_i \left( \boldsymbol{W}^{(r,k)} \right)^T$ is the head specific intermediate representation of node $i$ under relation $r$.

It might be interesting to consider cases where there are a different number of heads for different relationship types, as well as when a mixture of ARGAT and WIRGAT produce the attention coefficients, however, we leave that subject for future investigation and will not consider it further.

**Basis decomposition**  The number of parameters in the RGAT layer increases linearly with the number of relations $R$ and heads $K$, and can lead quickly to overparameterisation. In RGCNs it was found that decomposing the kernels was beneficial for generalisation, although it comes at the cost of increased model bias (Schlichtkrull et al., 2018). We follow this approach, decomposing both the kernels $\boldsymbol{W}^{(r,k)}$ as well as the kernels of attention mechanism $\boldsymbol{A}^{(r,k)}$ into $B_V$ basis matrices $\boldsymbol{V}^{(b)} \in \mathbb{R}^{F \times F'}$ and $B_X$ basis vectors $\boldsymbol{X}^{(b)} \in \mathbb{R}^{2F' \times D}$

$$\boldsymbol{W}^{(r,k)} = \sum_{b=1}^{B_W} c_b^{(r,k)} \, \boldsymbol{V}^{(b)}, \qquad\qquad \boldsymbol{A}^{(r,k)} = \sum_{b=1}^{B_X} d_b^{(r,k)} \, \boldsymbol{X}^{(b)}, \tag{10}$$

where $c_b^{(r,k)}, d_b^{(r,k)} \in \mathbb{R}$ are basis coefficients. We consider models using full and decomposed $\boldsymbol{W}$ and $\boldsymbol{A}$.

## 2.2  NODE CLASSIFICATION

For the transductive task of semi-supervised node classification, we employ a two-layer RGAT architecture shown in Figure 3a. We use a Rectified Linear Unit (RELU) activation after the RGAT concat layer, and a node-wise softmax on the final layer to produce an estimate for the probability that the $i^{\text{th}}$ label is in the class $\alpha$

$$P(\text{class}_i = \alpha) \approx \hat{y}_{i,\alpha} = \text{softmax}(\boldsymbol{h}_i^{(2)})_\alpha. \tag{11}$$

Table 1: A summary of the datasets used in our experiments and how they are partitioned.

| Datasets | AIFB | MUTAG | Tox21 |
|---|---|---|---|
| Task | Transductive | Transductive | Inductive |
| Nodes | 8,285 (1 graph) | 23,644 (1 graph) | 145,459 (8014 graphs) |
| Edges | 29,043 | 74,227 | 151,095 |
| Relations | 45 | 23 | 4 |
| Labelled | 176 | 340 | 96,168 (12 per graph) |
| Classes | 4 | 2 | 12 (multi-label) |
| Train nodes | 112 | 218 | (6411 graphs) |
| Validation nodes | 28 | 54 | (801 graphs) |
| Test nodes | 28 | 54 | (802 graphs) |

We then employ a masked cross-entropy loss $\mathcal{L}$ to constrain the network updates to the subset of nodes $\mathcal{Y}$ whose labels are known

$$\mathcal{L} = -\sum_{i \in \mathcal{Y}} \sum_{\alpha=1}^{n_{\text{classes}}} y_{i,\alpha} \ln\left(\hat{y}_{i,\alpha}\right), \tag{12}$$

where $\boldsymbol{y}_i$ is the one-hot representation of the true label for node $i$.

## 2.3 Graph classification

For inductive graph classification, we employ a two-layer RGAT followed by a graph gather and dense network architecture shown in Figure 3b. We use ReLU activations after each RGAT layer and the first dense layer. We use a tanh activation after the GraphGather : $\mathbb{R}^{N \times F} \to \mathbb{R}^{2F}$, which is a vector concatenation of the mean of the node representations with the feature-wise max of the node representations

$$\boldsymbol{H}' = \text{GraphGather}(\boldsymbol{H}) = \left(\frac{1}{N} \sum_{i=1}^{N} \boldsymbol{h}_i\right) \oplus \left[\bigoplus_{f=1}^{F} \max_i h_{i,f}\right]. \tag{13}$$

The final dense layer then produces logits of the size $n_{\text{classes}} \times n_{\text{tasks}}$, and we apply a task-wise softmax to its output to produce an estimate $\hat{y}_{t,\alpha}$ for the probability that the graph is in class $\alpha$ for a given task $t$, analogous to Equation (11). Weighted cross-entropy loss $\mathcal{L}$ is then used to form the learning objective

$$\mathcal{L}(w, y, \hat{y}) = -\sum_{t=1}^{n_{\text{tasks}}} \sum_{\alpha=1}^{n_{\text{classes}}} w_{t,\alpha} \, y_{t,\alpha} \ln\left(\hat{y}_{t,\alpha}\right), \tag{14}$$

where $w_{t,\alpha}$ and $y_{t,\alpha}$ are the weights and one-hot true labels for task $t$ and class $\alpha$ respectively.

## 3 Evaluation

### 3.1 Datasets

We evaluate the models on transductive and inductive tasks. Following the experimental setup of Schlichtkrull et al. (2018) for the transductive tasks, we evaluate our model on the Resource Description Framework (RDF) datasets AIFB and MUTAG. We also evaluate our model for an inductive task on the molecular dataset, Tox21. Details of these data sets are given in Table 1. For further details on the transductive and inductive datasets, please see Ristoski and Paulheim (2016) and Wu et al. (2018) respectively.

**Transductive baselines**  We consider as a baseline the recent state-of-the-art results from Schlichtkrull et al. (2018) obtained with a two-layer RGCN model with 16 hidden units and basis function decomposition. We also include the same challenging baselines of FEAT (Paulheim and Fümkranz, 2012), WL (Shervashidze et al., 2011; de Vries and de Rooij, 2015) and RDF2Vec (Ristoski and Paulheim, 2016). In-depth details of these baselines are given by Ristoski and Paulheim (2016).

**Inductive baselines**   As baselines for Tox21, we compare against the most competitive methods on Tox21 reported in Wu et al. (2018). Specifically, we compare against deep multitask networks Ramsundar et al. (2015), deep bypass multitask networks Wu et al. (2018), Weave Kearnes et al. (2016), and a RGCN model whose relational structure is determined by the degree of the node to be updated Altae-Tran et al. (2016). Specifically, up to and including some maximum degree $D_{\max}$,

$$\boldsymbol{h}'_i = \sigma \left[ (\boldsymbol{W}^{\deg(i)})^T \boldsymbol{h}_i + \sum_{j \in n_i} (\boldsymbol{U}^{\deg(i)})^T \boldsymbol{h}_j + \boldsymbol{b}^{\deg(i)} \right], \qquad (15)$$

where $\boldsymbol{W}^{\deg(i)} \in \mathbb{R}^{F \times F'}$ is a degree-specific linear transformation for self-connections, $\boldsymbol{U}^{\deg(i)} \in \mathbb{R}^{F \times F'}$ is a degree-specific linear transformation for neighbours into their intermediate representations $\boldsymbol{g}_i \in \mathbb{R}^{F'}$, and $\boldsymbol{b}^{\deg(i)}$ is a degree-specific bias. Any update for any degree $d(i) > D_{\max}$ gets assigned to the update for the maximum degree $D_{\max}$.

## 3.2   EXPERIMENTAL SETUP

**Transductive learning**   For the transductive learning tasks, the architecture discussed in Section 2.2 was applied. Its hyperparameters were optimised for both AIFB and MUTAG on their respective training/validation sets defined in Ristoski and Paulheim (2016), using 5-fold cross validation. Using the found hyperparameters, we retrain on the full training set and report results on the test set across 200 seeds. We employ early stopping on the validation set during cross-validation to determine the number of epochs we will run on the final training set. Hyperparameter optimisation details are given in Table 4 of Appendix B.

**Inductive learning**   For the inductive learning tasks, the architecture discussed in Section 2.3 was applied. In order to optimise hyperparameters once, ensure no data leakage, but also provide comparable benchmarks to those presented in Wu et al. (2018), three benchmark splits were taken from the `MolNet benchmarks`[4], and graphs belonging to any of the test sets were isolated. Using the remaining graphs we performed a hyperparameter search using 2 folds of 10-fold cross validation. Using the found hyperparameters, we then retrained on the three benchmark splits provided with 2 seeds each, giving an unbiased estimate of model performance. We employ early stopping during both the cross-validation and final run (the validation set of the inductive task is available for the final benchmark, in contrast to the transductive tasks) to determine the number of training epochs. Hyperparameter optimisation details are given in Table 5 of Appendix B.

**Constant attention**   In all experiments, we train with the attention mechanism turned on. At evaluation time, however, we report results with and without the attention mechanism to provide insight into whether the attention mechanism helps. ARGAT (WIRGAT) without the attention is called C-ARGAT (C-WIRGAT).

## 3.3   RESULTS

### 3.3.1   BENCHMARKS AND ADDITIONAL ANALYSES

Model means and standard deviations are presented in Table 2. To provide a picture of characteristic model behaviour, the CDFs for the hyperparameter sweep are presented in Figure 5 of Appendix D. To draw meaningful conclusions, we compare against our own implementation of RGCN rather than the results reported in Schlichtkrull et al. (2018); Wu et al. (2018).

We will occasionally employ a one-sided hypothesis test in order to make concrete statements about model performance. The details and complete results of this test are presented in Appendix E. When we refer to significant results this corresponds to a test statistic supporting our hypothesis with a $p-$value $p \le 0.05$.

---

[4]Retrieved   from   http://deepchem.io.s3-website-us-west-1.amazonaws.com/trained_models/Hyperparameter_MoleculeNetv3.tar.gz.

Table 2: (a) Entity classification results accuracy (mean and standard deviation over 10 seeds) for FEAT (Paulheim and Fümkranz, 2012), WL (Shervashidze et al., 2011; de Vries and de Rooij, 2015), RDF2Vec (Ristoski and Paulheim, 2016) and RGCN (Schlichtkrull et al., 2018), and (mean and standard deviation over 200 seeds) for our implementation of RGCN, as well as additive and multiplicative attention for (C-)WIRGAT and (C-)WIRGAT (this work). Test performance is reported on the splits provided in Ristoski and Paulheim (2016). (b) Graph classification mean Receiver Operating Characteristic (ROC) Area Under the Curve (AUC) across all 12 tasks (mean and standard deviation over 3 splits) for Multitask (Ramsundar et al., 2015), Bypass (Wu et al., 2018), Weave (Kearnes et al., 2016), RGCN (Altae-Tran et al., 2016), and (mean and standard deviation over 3 splits, 2 seeds per split) our implementation of RGCN, additive and multiplicative attention for (C-)WIRGAT and (C-)ARGAT (this work). Test performance is reported on the splits provided in Wu et al. (2018). Best performance in class in boldened, and best performance overall is underlined. For completeness, we present the training and validation mean ROC-AUCs alongside the test ROC-AUCs in Appendix A. For a graphical representation of these results, see Figure 4 in Appendix C.

| Model | AIFB | MUTAG |
|---|---|---|
| Feat | $55.55 \pm 0.00$ | $77.94 \pm 0.00$ |
| WL | $80.55 \pm 0.00$ | $\mathbf{80.88} \pm 0.00$ |
| RDF2Vec | $88.88 \pm 0.00$ | $67.20 \pm 1.24$ |
| RGCN | $95.83 \pm 0.62$ | $73.23 \pm 0.48$ |
| RGCN (ours) | $\mathbf{94.64} \pm 2.75$ | $74.15 \pm 2.40$ |
| *Additive attention* | | |
| C-WIRGAT | $\underline{\mathbf{96.86}} \pm 0.94$ | $69.37 \pm 2.75$ |
| WIRGAT | $96.83 \pm 1.01$ | $\mathbf{69.83} \pm 2.74$ |
| C-ARGAT | $93.05 \pm 3.05$ | $63.69 \pm 8.41$ |
| ARGAT | $94.01 \pm 2.76$ | $65.54 \pm 6.25$ |
| *Multiplicative attention* | | |
| C-WIRGAT | $93.71 \pm 3.33$ | $69.57 \pm 3.70$ |
| WIRGAT | $92.92 \pm 3.75$ | $69.60 \pm 3.75$ |
| C-ARGAT | $95.89 \pm 1.93$ | $\mathbf{74.38} \pm 3.78$ |
| ARGAT | $\mathbf{96.19} \pm 1.70$ | $73.17 \pm 3.41$ |

(a) Transductive

| Model | Tox21 |
|---|---|
| Multitask | $0.803 \pm 0.012$ |
| Bypass | $0.810 \pm 0.013$ |
| Weave | $0.820 \pm 0.010$ |
| RGCN | $0.829 \pm 0.006$ |
| RGCN (ours) | $\mathbf{0.835} \pm 0.008$ |
| *Additive attention* | |
| C-WIRGAT | $0.832 \pm 0.009$ |
| WIRGAT | $\mathbf{0.835} \pm 0.006$ |
| C-ARGAT | $0.829 \pm 0.010$ |
| ARGAT | $\mathbf{0.835} \pm 0.006$ |
| *Multiplicative attention* | |
| C-WIRGAT | $0.811 \pm 0.008$ |
| WIRGAT | $\underline{\mathbf{0.838}} \pm 0.007$ |
| C-ARGAT | $0.802 \pm 0.007$ |
| ARGAT | $0.837 \pm 0.007$ |

(b) Inductive

### 3.3.2 Transductive learning

In Table 2a we evaluate RGAT on MUTAG and AIFB. With additive attention, WIRGAT outperforms ARGAT, consistent with Schlichtkrull et al. (2018). Interestingly, when employing multiplicative attention, the converse appears true. For node classification tasks on RDF data, this indicates that the importance of a particular relation type does not vary much (if at all) across the graph unless one employs a multiplicative comparison[5] between node representations.

**AIFB** On AIFB, the best to worst performing models are: 1) additive WIRGAT 2) multiplicative ARGAT 3) RGCN 4) additive ARGAT, and 5) multiplicative WIRGAT, with each comparison being significant.

When comparing against their constant attention counterparts, the significant differences observed were for additive and multiplicative ARGAT, where attention gives a relative mean performance improvements of 1.03% and 0.31% respectively, and multiplicative WIRGAT, where attention gives a relative mean performance drop of 0.84%.

---

[5]Or potentially other comparisons beyond additive or constant, i.e. RGCN.

Although we present state-of-the art result on AIFB with additive WIRGAT, since its performance with and without attention are not significantly different, it is unlikely that this is due to the attention mechanism itself, at least at inference time. Over the hyperparameter space, additive WIRGAT and RGCN are comparable in performance (see Figure 5a in Appendix D), leading us to conclude that the result is more likely attributable to finding a better hyperparameter point for additive WIRGAT during the search.

**MUTAG** On MUTAG, the best to worst performing models are: 1) RGCN 2) multiplicative ARGAT 3) additive WIRGAT tied with multiplicative WIRGAT, and 4) additive ARGAT, with each comparison being significant.

When comparing against their constant attention counterparts, the significant differences observed were for additive WIRGAT and ARGAT, where attention gives relative mean performance improvements of 0.66% and 2.90% respectively, and multiplicative ARGAT, where attention gives a relative mean performance drop of 1.63%.

We note that RGCN consistently outperforms RGAT on MUTAG, contrary to what might be expected (Schlichtkrull et al., 2018). The result is surprising given that RGCN lies within the parameter space of RGAT (where the attention kernel is zero), a configuration we check through evaluating C-WIRGAT. In our experiments we have observed that both RGCN and RGAT can memorise the MUTAG training set with 100% accuracy without difficulty (this is not the case for AIFB). The performance gap between RGCN and RGAT could then be explained by the following:

- During training, the RGAT layer uses its attention mechanism to solve the learning objective. Once the objective is solved, the model is not encouraged by the loss function to seek a point in the parameter space that would also behave well when attention is set to a normalising constant within neighbourhoods (i.e. the parameter space point that would be found by RGCN).

- The RDF tasks are transductive, meaning that a basis-dependent spectral approach is sufficient to solve them. As RGCN already memorises the MUTAG training set, a model more complex[6] than RGCN, for example RGAT, that can also memorise the training set is unlikely to generalise as well, although this is a hotly debated topic - see e.g. Zhang et al. (2016).

We employed a suite of regularisation techniques to get RGAT to generalise on MUTAG, including L2-norm penalties, dropout in multiple places, batch normalisation, parameter reduction and early stopping, however, no evaluated harshly regularised points for RGAT generalise well on MUTAG.

Our final observation is that the attention mechanism presented in Section 2.1 relies on node features. The node features for the above tasks are learned from scratch (the input feature matrix is a one-hot node index) as part of the task. It is possible that in this semi-supervised setup, there is insufficient signal in the data to learn both the input node embeddings as well as a meaningful attention mechanism to act upon them.

### 3.3.3 Inductive learning

In Table 2b we evaluate RGAT on Tox21. The number of samples is lower for these evaluations than for the transductive tasks, and so fewer model comparisons will be accompanied with a reasonable significance, although there are still some conclusions we can draw.

Through a thorough hyperparameter search, and incorporating various regularisation techniques, we obtained the relative mean performance of 0.72% for RGCN compared to the result reported in Wu et al. (2018), providing a much stronger baseline.

Both additive attention models match the performance of RGCN, whereas multiplicative WIRGAT and ARGAT marginally outperform RGCN, although this is not significant ($p = 0.24$ and $p = 0.41$ respectively).

---

[6]Measured in terms of Minimum Description Length (MDL), for example.

When comparing against their constant attention counterparts, significant differences observed were for multiplicative WIRGAT and ARGAT, where attention gives a relative mean performance improvements of 3.33% and 4.36% respectively. We do not observe any significant gains coming from additive attention when compared to their constant counterparts.

## 4  CONCLUSION

We have investigated a class of models we call Relational Graph Attention Networks (RGATs). These models act upon graph structures, inducing a masked self-attention that takes account of local relational structure as well as node features. This allows both nodes and their properties under specific relations to be dynamically assigned an importance for different nodes in the graph, and opens up graph attention mechanisms to a wider variety of problems.

We evaluted two specific attention mechanisms, Within-Relation Graph Attention (WIRGAT) and Across-Relation Graph Attention (ARGAT), under both an additive and multiplicative logit construction, and compared them to their equivalently evaluated spectral counterpart Relational Graph Convolutional Networks (RGCNs).

We find RGATs perform competitively or poorly on established baselines. This behavior appears strongly task-dependent. Specifically, relational inductive tasks such as graph classification benefit from multiplicative ARGAT, whereas transductive relational tasks, such as knowledge base completion, at least in the absence of node features, are better tackled using spectral methods like RGCNs or other graph feature extraction methods like Weisfeiler-Lehman (WL) graph kernels.

In general we have found that WIRGAT should be paired with an additive logit mechanism, and fares marginally better than ARGAT on transductive tasks, whereas ARGAT should be paired with a multiplicative logit mechanism, and fares marginally better on inductive tasks.

We have found no cases where choosing any variation of RGAT is guaranteed to significantly outperform RGCN, although we have found that in cases where RGCN can memorise the training set, we are confident that RGAT will not perform as well as RGCN. Consequently, we suggest that before attempting to train RGAT, a good first test is to inspect the training set performance of RGCN.

Through our thorough evaluation and presentation of the behaviours and limitations of these models, insights can be derived that will enable the discovery of more powerful model architectures that act upon relational structures. Observing that model variance on all of the tasks presented here is high, any future work developing and expanding these methods must choose larger, more challenging datasets. In addition, a comparison between the generalisation of spectral methods, like those presented here, and generalisations of Recurrent Neural Networks (RNNs), like Gated Graph Sequence Networks, is a necessary ingredient for determining the most promising future direction for these models.

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

## A  Tox21 Results

For completeness, we present the training, validation and test set performance of our models in addition to those in Wu et al. (2018) in Table 3.

Table 3: Graph classification mean Area Under the Curve (AUC) across all 12 tasks (mean and standard deviation over 3 splits) for Multitask (Ramsundar et al., 2015), Bypass (Wu et al., 2018), Weave (Kearnes et al., 2016), RGCN (Altae-Tran et al., 2016), our implementation of RGCN, additive and multiplicative attention versions of WIRGAT and ARGAT (this work). Training, validation and test performance is reported on the splits provided in Wu et al. (2018). Best performance in class in boldened, and best performance overall is underlined.

| Model | Training | Validation | Test |
|---|---|---|---|
| Multitask | $0.884 \pm 0.001$ | $0.795 \pm 0.017$ | $0.803 \pm 0.012$ |
| Bypass | $0.938 \pm 0.001$ | $0.800 \pm 0.008$ | $0.810 \pm 0.013$ |
| Weave | $0.875 \pm 0.004$ | $0.828 \pm 0.008$ | $0.820 \pm 0.010$ |
| RGCN | $\mathbf{0.905} \pm 0.004$ | $0.825 \pm 0.013$ | $0.829 \pm 0.006$ |
| RGCN (ours) | $0.883 \pm 0.010$ | $\mathbf{0.845} \pm 0.003$ | $\mathbf{0.835} \pm 0.008$ |
| *Additive attention* | | | |
| C-WIRGAT | $0.897 \pm 0.022$ | $0.842 \pm 0.004$ | $0.832 \pm 0.009$ |
| WIRGAT | $\mathbf{0.902} \pm 0.024$ | $0.845 \pm 0.005$ | $\mathbf{0.835} \pm 0.006$ |
| C-ARGAT | $0.884 \pm 0.012$ | $0.848 \pm 0.003$ | $0.829 \pm 0.010$ |
| ARGAT | $0.896 \pm 0.016$ | $\mathbf{0.851} \pm 0.004$ | $\mathbf{0.835} \pm 0.006$ |
| *Multiplicative attention* | | | |
| C-WIRGAT | $0.859 \pm 0.016$ | $0.830 \pm 0.007$ | $0.811 \pm 0.008$ |
| WIRGAT | $\mathbf{0.904} \pm 0.022$ | $\underline{\mathbf{0.852}} \pm 0.002$ | $\underline{\mathbf{0.838}} \pm 0.007$ |
| C-ARGAT | $0.838 \pm 0.007$ | $0.816 \pm 0.007$ | $0.802 \pm 0.007$ |
| ARGAT | $0.802 \pm 0.007$ | $0.846 \pm 0.003$ | $0.837 \pm 0.007$ |

## B  Hyperparameters

We perform hyperparameter optimisation using `hyperopt` Bergstra et al. (2013) with priors for the transductive tasks specified in Table 4 and priors for the inductive tasks specified in Table 5. In all experiments we use the Adam optimiser (Kingma and Ba, 2014).

Table 4: Priors on the hyperparameter search space for the transductive tasks. When multihead attention is used, the number of units per head is appropriately reduced in order to keep the total number of output units of an RGAT layer independent of the number of heads.

| Hyperparameter | Prior |
|---|---|
| Graph kernel units | $\mathrm{MultiplesOfFour}(4, 20)$ |
| Heads | $\mathrm{OneOf}(1, 2, 4)$ |
| Feature dropout rate | $\mathrm{Uniform}(0.0, 0.8)$ |
| Edge dropout | $\mathrm{Uniform}(0.0, 0.8)$ |
| $\boldsymbol{W}$ basis size | $\mathrm{OneOf}(\mathrm{Full}, 5, 10, 20, 30)$ |
| Graph layer 1 $\boldsymbol{W}$ L2 coef | $\mathrm{LogUniform}(10^{-6}, 10^{-1})$ |
| Graph layer 2 $\boldsymbol{W}$ L2 coef | $\mathrm{LogUniform}(10^{-6}, 10^{-1})$ |
| $\boldsymbol{A}$ basis size | $\mathrm{OneOf}(\mathrm{Full}, 5, 10, 20, 30)$ |
| Graph layer 1 $\boldsymbol{A}$ L2 coef | $\mathrm{LogUniform}(10^{-6}, 10^{-1})$ |
| Graph layer 2 $\boldsymbol{A}$ L2 coef | $\mathrm{LogUniform}(10^{-6}, 10^{-1})$ |
| Learning rate | $\mathrm{LogUniform}(10^{-5}, 10^{-1})$ |
| Use bias | $\mathrm{OneOf}(\mathrm{Yes}, \mathrm{No})$ |
| Use batch normalisation | $\mathrm{OneOf}(\mathrm{Yes}, \mathrm{No})$ |

Table 5: Priors on the hyperparameter for the inductive task. The batch size was held at 64, and no bases decomposition is used. When multihead attention is used, the number of units per head is appropriately reduced in order to keep the total number of output units of an RGAT layer independent of the number of heads.

| Hyperparameter | Prior |
|---|---|
| Graph kernel units | $\mathrm{MultiplesOfEight}(32, 128)$ |
| Dense units | $\mathrm{MultiplesOfEight}(32, 128)$ |
| Heads | $\mathrm{OneOf}(1, 2, 4, 8)$ |
| Feature dropout | $\mathrm{Uniform}(0.0, 0.8)$ |
| Edge dropout | $\mathrm{Uniform}(0.0, 0.8)$ |
| $\boldsymbol{W}$ L2 coef (1) | $\mathrm{LogUniform}(10^{-6}, 10^{-1})$ |
| $\boldsymbol{W}$ L2 coef (2) | $\mathrm{LogUniform}(10^{-6}, 10^{-1})$ |
| $\boldsymbol{a}$ L2 coef (1) | $\mathrm{LogUniform}(10^{-6}, 10^{-1})$ |
| $\boldsymbol{a}$ L2 coef (2) | $\mathrm{LogUniform}(10^{-6}, 10^{-1})$ |
| Learning rate | $\mathrm{LogUniform}(10^{-5}, 10^{-1})$ |
| Use bias | $\mathrm{OneOf}(\mathrm{Yes}, \mathrm{No})$ |
| Use batch normalisation | $\mathrm{OneOf}(\mathrm{Yes}, \mathrm{No})$ |

## C   CHARTS

To aid interpretability of the results presented in Table 2 we present a chart representation in Figure 4.

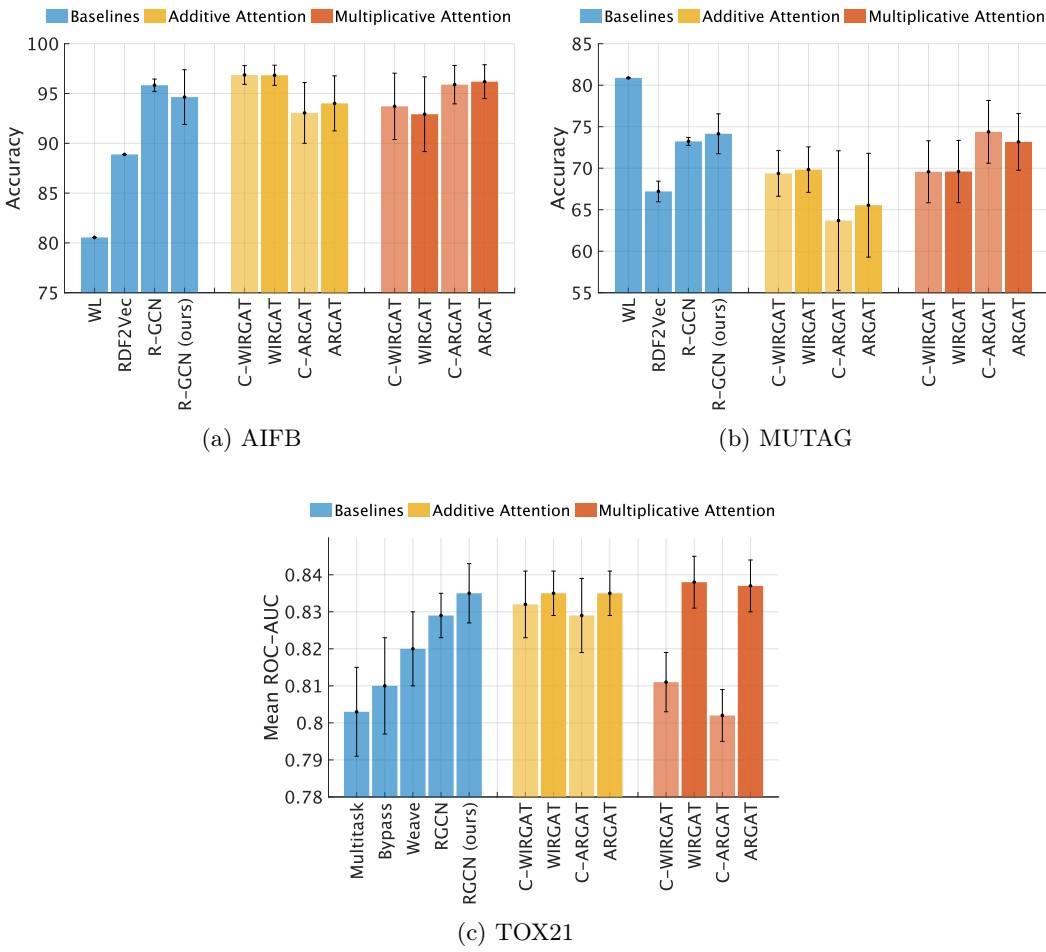

(a) AIFB

(b) MUTAG

(c) TOX21

Figure 4: (a) and (b): **Blue** Baseline entity classification accuracy (mean and standard deviation over 10 seeds) for FEAT (Paulheim and Fümkranz, 2012), WL (Shervashidze et al., 2011; de Vries and de Rooij, 2015), RDF2Vec (Ristoski and Paulheim, 2016) and RGCN (Schlichtkrull et al., 2018), and (mean and standard deviation over 200 runs) for our implementation of RGCN. **Yellow** Entity classification accuracy (mean and standard deviation over 200 seeds) for additive attention (this work). **Red** Entity classification accuracy (mean and standard deviation over 200 seeds) for multiplicative attention (this work). Test performance is reported on the splits provided in Ristoski and Paulheim (2016). (c): **Blue** Baseline graph classification mean Receiver Operating Characteristic (ROC) AUC across all 12 tasks (mean and standard deviation over 3 splits) for Multitask (Ramsundar et al., 2015), Bypass (Wu et al., 2018), Weave (Kearnes et al., 2016), RGCN (Altae-Tran et al., 2016), and (mean and standard deviation over 3 splits, 2 seeds per split) our implementation of RGCN. **Yellow** Additive attention graph classification mean ROC-AUC (mean and standard deviation over 200 seeds) across all 12 tasks (this work). **Red** Multiplicative attention graph classification mean ROC-AUC (mean and standard deviation over 200 seeds) across all 12 tasks (this work). All raw values are given in Table 2.

# D  CUMULATIVE DISTRIBUTION FUNCTIONS

To aid further insight into our results, we present the Cumulative Distribution Functions (CDFs) for each model on each task in Figure 4. In this context, we treat the performance metric of interest during the hyperparameter search as the empirical distribution of some random variable $X$. We then define its CDF $F_X(x)$ in the standard way

$$F_X(x) = P(X \leq x), \tag{16}$$

where $P(X \leq x)$ is the probability that $X$ takes on a value less than or equal to $x$. The CDF allows one to gauge whether any given architecture typically performs better than another across the whole space, rather than comparison of the tuned hyperparameters, which in some cases may be outliers in terms of generic behavior for that architecture.

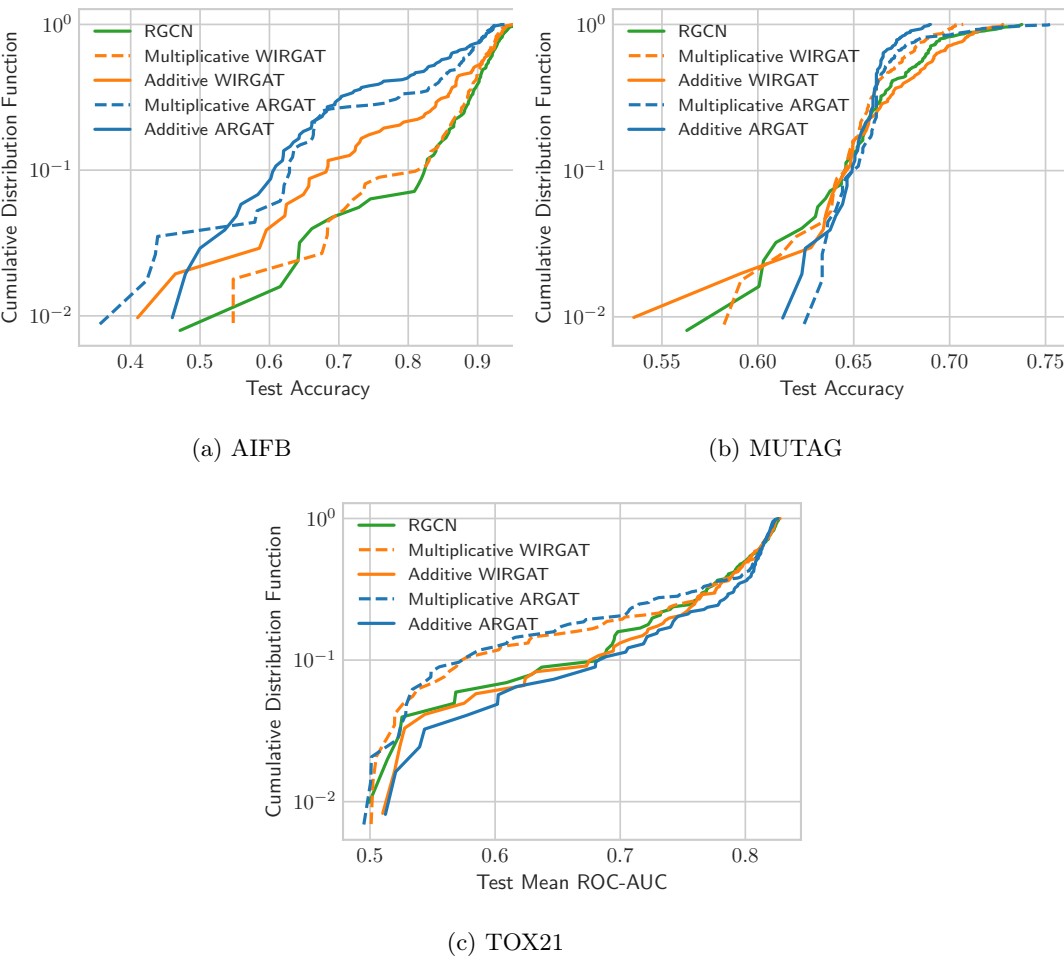

(a) AIFB                                      (b) MUTAG

(c) TOX21

Figure 5: CDFs for all models on a) AIFB, b) MUTAG and c) TOX21. Green lines correspond to our implementation of RGCN, blue lines correspond to ARGAT, and orange lines correspond to WIRGAT. Solid lines correspond to additive attention (and RGCN), whereas dashed lines correspond to multiplicative attention. A lower CDF value is better in the sense that a greater proportion of models of achieve a higher value of that metric.

**AIFB**  Additive and multiplicative ARGAT perform poorly for most areas of the hyperparameter space, whereas RGCN and multiplicative WIRGAT perform comparably across the entire hyperparameter space.

**MUTAG**   Interestingly, the models that have a greater amount of hyperparameter space covering poor performance (i.e. RGCN, multiplicative and additive WIRGAT) are also the models which also have a greater amount of hyperparameter space covering good performance. In other words, on the MUTAG, the ARGAT prior resulted in a model whose test set performance was relatively insensitive to hyperparameter choice when compared against the other candidates. Given that the ARGAT model was the most flexible of the models evaluated, and that it was able to memorise the training set, this suggests that the task contained insufficient information for the model to learn its attention mechanism. Given that WIRGAT was able to at least partially learn to its attention mechanism suggests that WIRGAT is less data hungry than ARGAT.

**Tox21**   The multiplicative attention models fare poorly on the majority of the hyperparameter space compared to the other models. There is a slice of the hyperparameter space where the multiplicative attention models outperform the other models, however, indicating that although they are difficult to train, it may be worth spending time hyperoptimising them if you need the best performing model on a relational inductive task. The additive attention models and RGCN perform comparably across the entirety of the hyperparameter space and generally perform better than the multiplicative methods except for the very small region of hyperparameter space mentioned above.

# E  Significance testing

In order to determine if any of our model comparisons are significant, we employ the one-sided Mann-Whitney $U$ test Mann and Whitney (1947) as we are interested in the direction of movement (i.e. performance) and do not want to make any parametric assumptions about model response. For two populations $X$ and $Y$:

- The null hypothesis $H_0$ is that the two populations are equal, and
- The alternative hypothesis $H_1$ is that the probability of an observation from population X exceeding an observation from population Y is larger than the probability of an observation from Y exceeding an observation from X; i.e., $H_1 : P(X > Y) > P(Y > X)$.

We treat the empirical distributions of Model A as samples from population $X$ and the empirical distributions of Model B as samples from population $Y$. This allows us a window into whether, given a task, whether which is the better model out of a pair of models. Results on AIFB, MUTAG and TOX21 are given in Figure 6, Figure 7 and Figure 8 respectively.

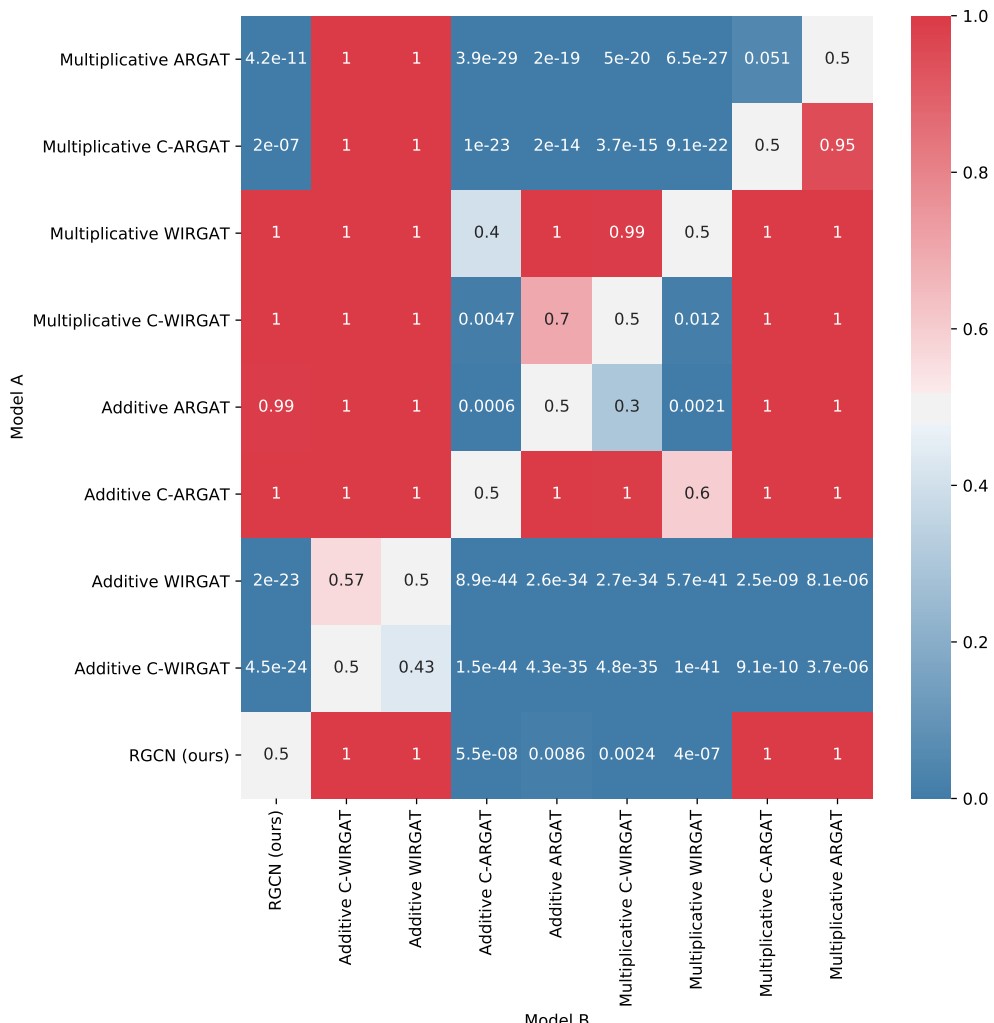

Figure 6: The $p$-values for Mann-Whitney $U$ test with alternative hypothesis $H_1$ of Model $A$ outperforming Model $B$ on AIFB.

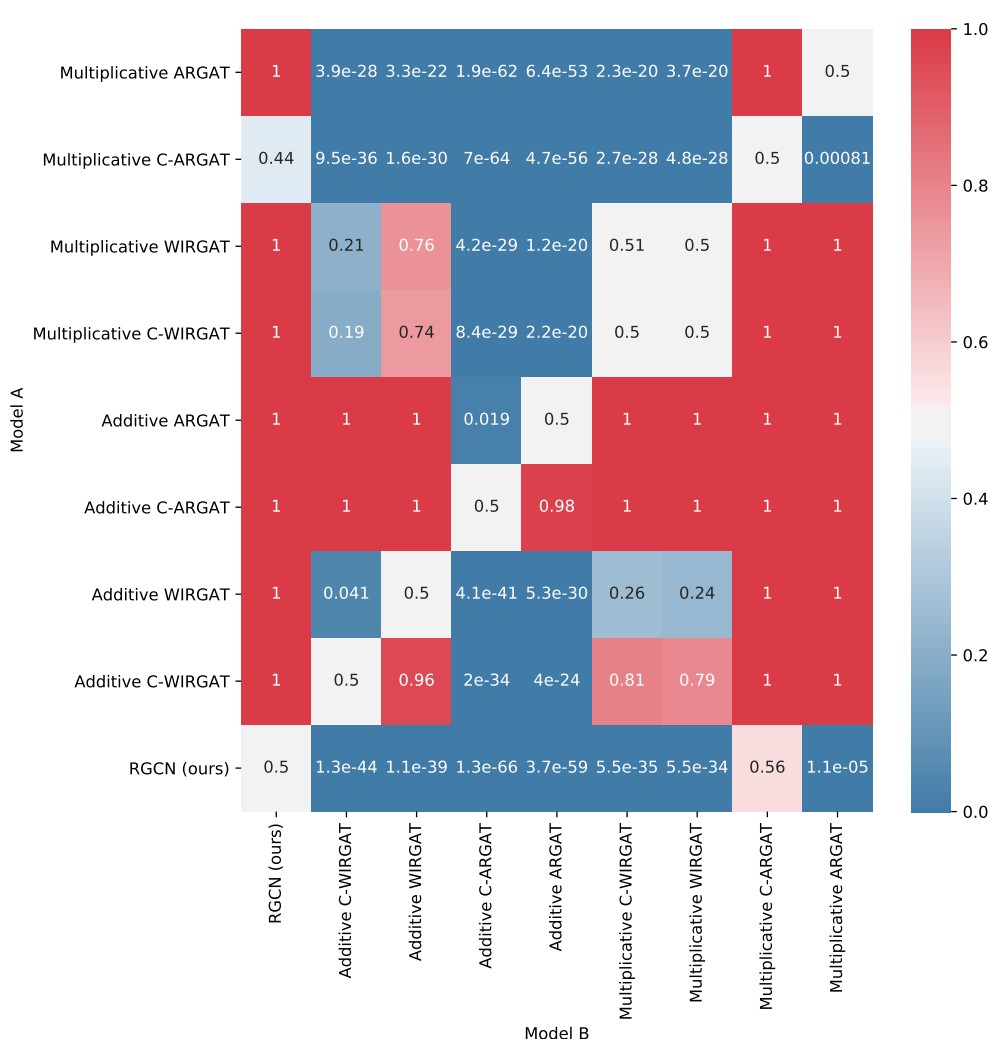

Figure 7: The $p$-values for Mann-Whitney $U$ test with alternative hypothesis $H_1$ of Model $A$ outperforming Model $B$ on MUTAG.

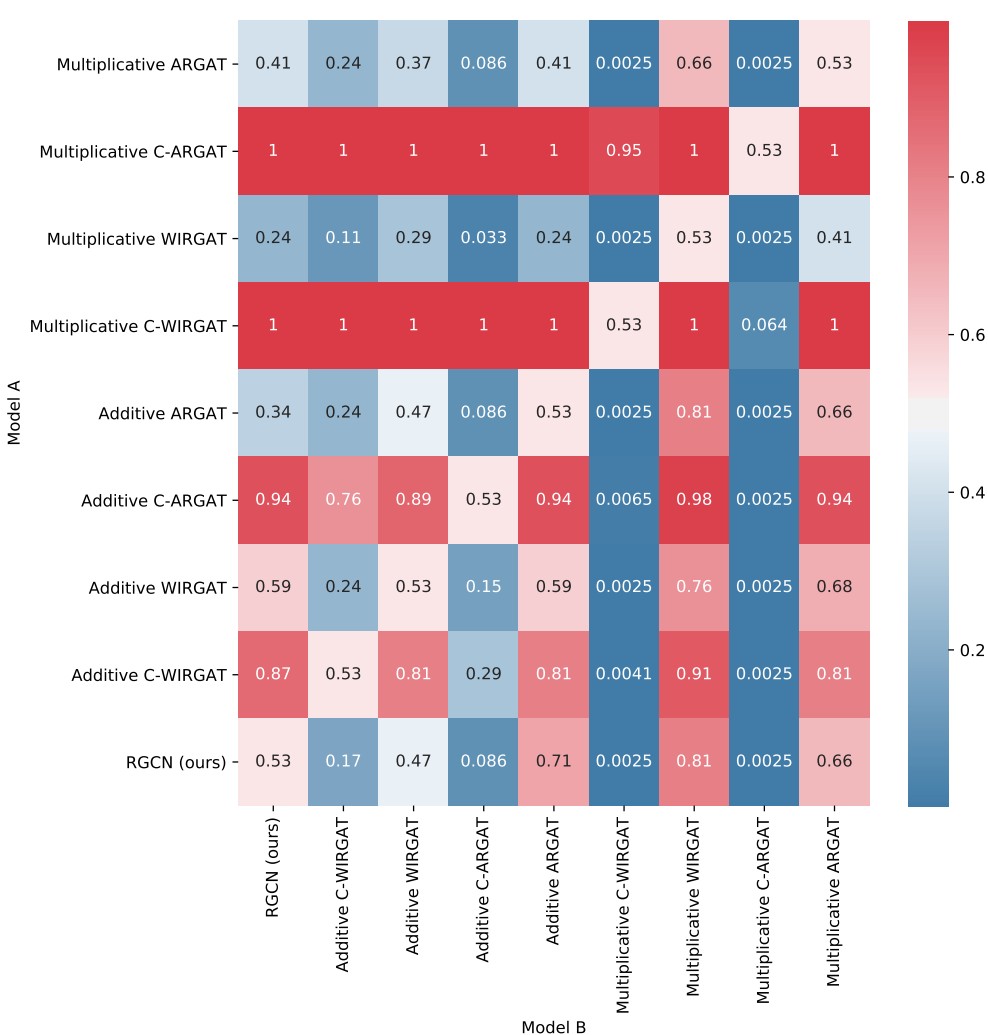

Figure 8: The $p$-values for Mann-Whitney $U$ test with alternative hypothesis $H_1$ of Model $A$ outperforming Model $B$ on TOX21.

