# OpenReview forum: "Relational Graph Attention Networks"
_ICLR.cc/2019/Conference_

### Official Review · AnonReviewer1 · 2018-11-01
**A slight reformulation of RGCN with attention mechanisms with mixed results on graph classification and node classification tasks.**

**Rating:** 4
**Confidence:** 5

**Review:**

The paper proposes a few variations on the RGCN model with adding attention to either within relations or across relations.

Unfortunately the paper falls short in two main areas:

- novelty: the additions proposed are small modifications to existing algorithms and there are other methods of attention on graphs which have been discussed in the paper but not directly compared to (e.g. this method builds heavily on Velickovic et al 2017)
- impact: the results achieved in the experiments are very small improvements compared to the baseline of RGCN (~ +0.01 in two experiments and ~ -0.04 in another) and often these small variations in results can be compensated with better baselines training (e.g. better hyper-params, ...)

However, on a positive note, the paper has been written very well and I really liked the frank discussion on page 8 about results on MUTAG dataset.

---

> ### Author Response · Authors · 2018-11-15
> **Response to Reviewer 1**
>
> Dear Reviewer 1, thank you for taking time to read and review our paper and for your useful comments. Hopefully the new results in our response will better aid discussion. Your specific points are addressed below.
>
> > “...the additions proposed are small modifications to existing algorithms
>
> We concede that the modifications to the existing models is a minor contribution. We would like to highlight that despite being a simple modification, producing an implementation that trains in a reasonable about ot time is non-trivial - this is confirmed by community comments requesting for implementation details. We plan to make our code public to aid research in the area.
>
> To generalise the model, following the recommendations in one of the comments, we have investigated applying the Transformer-style dot product attention that was presented in GaAN (Zhang at al. 2018 https://arxiv.org/abs/1803.07294). This generalises the notion of RGAT, and we believe that this increase the contributions in terms of model modification that our paper offers.
>
> > “...and there are other methods of attention on graphs which have been discussed in the paper but not directly compared to (e.g. this method builds heavily on Velickovic et al 2017).”
>
> Unfortunately we cannot directly compare our approach to Velickovic et al. (2017) on relational graphs since their proposed model doesn’t support relationship types. Hence, we only compare to Schlichtkrull et al. (2017) and the conventional baselines, whereas Velickovic et al (2017) reports results on the non-relational graphs Cora, Citeseer, Pubmed and PPI.
>
> In the case of the RDF tasks, our model hyperparameter search space does include a model very similar to vanilla GAT. In the case where we only have both one basis kernel for each of the convolution and the attention, i.e. where B_V and B_v in equation (8) are set to 1, then the difference between RGAT and GAT is only a set of learnable relative scale factors living inside the basis coefficients c_b and c_d. During our hyperparameter search, however, no favourable points for evaluation set performance were discovered with basis sizes lower than 10 (a basis size of 5 is permitted in the search). This leads us to conclude that vanilla GAT would not perform well on the RDF tasks.
>
> As mentioned above, we have now also evaluated the dot-product style attention of the transformer and have included it in our results. As one of the public comments mentioned, a study comparing these types of attention models to the more recurrent based models like Gated Graph Neural Networks (Li et al. 2015) would be extremely worthwhile. We we feel that evaluation lies outside of the scope of this work, however, which is mainly concerned with evaluating how the introduction of an attention mechanism into RGCN modifies its behaviour.
>
> > “...the results achieved in the experiments are very small improvements compared to the baseline of RGCN…”
>
> We agree that any improvements compared to RGCN are marginal. In light of your third comment (below) regarding hyperparameters, the new RGCN baseline on Tox21 significantly closes the gap between the baseline and sum-attention RGAT. The new dot product attention results on Tox21 (Mean test AUCs: WIRGAT 0.838 +/- 0.007, ARGAT 0.837 +/- 0.007) are slightly improved compared those of the sum attention, however, due to the retrained RGCN baseline, the relative gap between the best performing RGAT and RGCN is now smaller than it was before.
>
> We also see value in reporting these negative results. It was expected that an attention mechanism like GAT should cope with the node-degree imbalances observed to be present in the MUTAG dataset [a statement along these lines was made in Schlichtkrull et al. (2017)]. The most natural route to tackle this problem turns out to fail - a result which we believe is informative for the community who are trying to solve this problem. On the other hand, the newly evaluated dot-product attention does no worse (or better) than RGCN, indicating a more promising research direction to pursue.
>
> > “...often these small variations in results can be compensated with better baselines training…”
>
> We also suspected this was a possibility, and in the same of sum-style attention it turns out to be true.
>
> To determine whether this was the case, we performed the same hyperparameter optimisations to our implementation of RGCN. In the case of the RDF datasets AIFB and MUTAG, we observe no meaningful difference between our retrained RGCN benchmark and the original benchmark provided in Schlichtkrull et al. (2017). On the other hand, our retrained implementation on the graph classification task TOX21 raised the performance above that of the RGCN reported in Wu et al, 2017 ti match the performance of the sum-style attention mechanism RGAT performance. This new RGCN performance is not higher than the observed performance of the dot-style attention mechanism, however, although this is not significant as discussed above.

---

### Official Review · AnonReviewer2 · 2018-11-03
**Small incremental extension of existing work**

**Rating:** 4
**Confidence:** 5

**Review:**

This work extends Schlichtkrull et al. (2018) by adding attention in two distinct ways: attention between pairs of nodes per relation, and attention between pairs of nodes averaged over all relations. The paper is well written and the equations easy to follow. The results are not strong. And, unfortunately, the model contribution currently is too modest.

Inductive task results: Wu et al. (2018) reports that for Tox21 (Duvenauld et al. 2015) is the best-performing approach. We should see the performance on other datasets  (e.g., some of the other datasets in Wu et al. (2018)).

My introduction suggestion: do not talk about Convolutional neural networks (CNNs). There is a *lot* of work on graph convolutional networks (GCNs). Reading it feels like reading about lattices when the work is about general graphs, and lattices provide no intuition about the proposed solution.

--- After rebuttal ---

Still not convinced of the value of the work to the community. Will keep my score the same.

---

> ### Author Response · Authors · 2018-11-15
> **Response to Reviewer 2**
>
> Dear Reviewer 2, thank you for your constructive feedback and for taking the time to review our work. Your specific points are addressed below.
>
> > “The results are not strong. And, unfortunately, the model contribution currently is too modest.”
>
> Indeed, the model is a minor contribution and, especially in light of a more thorough evaluation of RGCN, the sum attention RGAT results do not improve on those in Schlichtkrull et al. (2017). However, we would like to highlight that despite being a simple modification, the implementation of such a model that is trainable in a reasonable time frame is non-trivial. We plan to make our implementation public to aid future research in the area.
>
> To generalise the model, following the recommendations in one of the comments, we have investigated applying the Transformer-style dot product attention that was presented in GaAN: Gated Attention Networks for Learning on Large and Spatiotemporal Graphs (Zhang et al. 2018). We believe that this additional investigation, accompanied by marginally improved results on Tox21 for dot product attention (Mean test AUCs: WIRGAT 0.838 +/- 0.007, ARGAT 0.837 +/- 0.007) enhance model contributions of the paper.
>
> We also consider the poor results on MUTAG to be significant and informative for the rest of the community in developing more powerful models that can be applied to relational graphs.
>
> > “We should see the performance on other datasets  (e.g., some of the other datasets in Wu et al. (2018)”
>
> We agree that including experiments on the other data sets presented in Wu et al. (2018) would be a valuable addition to the paper. Unfortunately, we are unable to perform the required thorough hyperparameter exploration required to draw meaningful conclusions within the remaining time of the rebuttal period. This is something we will investigate in the future.
>
> > “My introduction suggestion: do not talk about Convolutional neural networks (CNNs).”
>
> Thank you for this stylistic critique. We see how the approach taken did not provide an intuition about the problem as well as it could have.
>
> We agree with your point regarding the wealth of graph neural network studies. We feel that the field of geometric deep learning, from which part of the direction of this work originated, is important to keep as part of the development of graph-based machine learning models. Some recent publications in the area of graph based ML have put less emphasis on geometric deep learning, the generalisations of convolutions from grids to graph, and the modifications of convolutions to achieve non-basis dependent methods. We felt that it is important to keep these concepts associated with the field of graph based ML. This introduction could, however, talk less in detail about CNNs themselves, and deal more with graphs - the main focus of the paper.
>
> We will produce a reworked introduction where graphs play a larger role. This should provide a more intuitive introduction to our work, whilst maintaining cornerstone concepts.

---

### Official Review · AnonReviewer3 · 2018-11-06
**A good submission but not good enough**

**Rating:** 4
**Confidence:** 4

**Review:**

This paper presented a relational graph attention networks that could consider both node
features and relational information (edge features) to perform node-level and graph-level
classifications. The basic idea is to combine the graph attention networks (Veličković et
al. 2017) and the relational graph networks (Schlichtkrull et al. 2018) to derive a hybrid
networks. This paper is generally easy to follow and written clearly. Several experiments
are conducted to demonstrate the performance of the proposed model. Although some promising
results have been achieved, I think there are several limitations regarding the novelty and
significance of the proposed model.

i) The proposed architecture is mainly adopted from the graph attention networks (Veličković
et al. 2017) and the relational graph networks (Schlichtkrull et al. 2018). Such a simple
combination is a good attempt to incorporate both node features and edge features but the
novelty is quite limited.


ii) In table 2, I don’t really see any promising results compared to baselines. There are
little improvements over the baselines or even significantly worse. More importantly,
compared two schemes of this work, the ones with attentions are “almost” identical with ones
without attentions, which implies that the proposed attentions mechanism is not really useful
in practice. For most of newly proposed graph embedding algorithms, it is hard to convince
it is indeed better without some significant improvements (at least 2% absolute accuracy more).

iii) For MUTAG dataset, the statistical information of this dataset is quite different from
what I used to use. MUTAG is a standard dataset for testing graph-level classification for
both graph kernels and graph networks. MUTAG is a dataset of 188 mutagenic aromatic and
heteroaromatic nitro compounds with 7 discrete node labels. Each chemical compound is labeled
according to whether it has mutagenic effect on the gram- negative bacterium Salmonnella
Typhimurium. Could you explain why your MUTAG is now a single graph and is cast as node
classification problem?

---

> ### Author Response · Authors · 2018-11-14
> **Response to Reviewer 3**
>
> Dear Reviewer 3, thank you for taking time to read and review our paper and for your useful comments. Hopefully the new results in our response will better aid discussion. Your specific points are addressed below.
>
> > i) “The proposed architecture is mainly adopted from the graph attention networks (Veličković  et al. 2017) and the relational graph networks (Schlichtkrull et al. 2018). Such a simple combination is a good attempt to incorporate both node features and edge features but the novelty is quite limited.”
>
> We concede that the modifications to the existing models is a minor contribution. We would like to highlight that despite being a simple modification, the implementation of such a model that is trainable in a reasonable time frame is non-trivial. We plan to make our implementation public to aid research in the area.
>
> To generalise the model, following the recommendations in one of the comments, we have investigated applying the Transformer-style dot product attention that was presented in GaAN: Gated Attention Networks for Learning on Large and Spatiotemporal Graphs (Zhang et al. 2018) for non-relational graphs. We believe that this additional investigation, accompanied by marginally improved results on Tox21 for dot product attention (Mean test AUCs: WIRGAT 0.838 +/- 0.007, ARGAT 0.837 +/- 0.007) enhance the paper’s contribution.
>
> > ii) “In table 2, I don’t really see any promising results compared to baselines. There are little improvements over the baselines or even significantly worse.”
>
> We agree that the results do not improve on those in Schlichtkrull et al. (2017) or Wu et al. (2017) in a significant way. This statement especially in light of the more thorough experiments conducted since - we have performed the same level of hyperparameter optimisation to an RGCN model and have found that the there is no gap between RGCN and sum-style attention RGAT. These results will be included in the new manuscript.
>
> We also feel that some of the results being “significantly worse” is one of the main contributions of our paper. Specifically, the results on MUTAG for RGAT with sum attention are quite a lot lower than its RGCN counterpart. This was an unexpected result for us as GAT-style attention was anticipated to handle the node-degree imbalances observed to be present in the MUTAG dataset [a statement along these lines was made in Schlichtkrull et al. (2017)]. The most natural route to tackle this problem turns out to fail - a result which we believe is informative for the community who are trying to solve this problem. On the other hand, the newly evaluated dot-product attention does no worse (or better) than RGCN, indicating a more promising research direction to pursue.
>
> > iii) “Could you explain why your MUTAG is now a single graph and is cast as node classification problem?”
>
> The MUTAG dataset in its standard form from e.g. TUD Benchmark Data Sets for Graph Kernels you are absolutely correct in your description.
>
> There is an alternative dataset version of mutag distributed with dl-learner [1] that is leveraged in the approach of RDF2Vec from Ristoski and Paulheim 2016 [2]. In this scenario, MUTAG is in presented in the RDF, where each of the 340 presented complex molecules is described by the links in an RDF with edges (triples) of the form
> d1 -> hasAtom -> d1_1
> d1 -> hasBond -> bond1
> d1- > hasStructure -> ring_size_6-1
> where in this case, d1 is the complex molecule, d1_1 is the first atom in complex molecule d1, and bond1 is the first bond in complex molecule d1, and so on. There are many more types than this, and are viewable in the .owl located at [2]. Nodes correspond to entities from the point of view of RDF. Each complex molecule is given a label according to whether it is mutagenic or not (determined from an isMutagenic property in the RDF - this is removed from the dataset before training/evaluation). This corresponds to 340 labels in total and constitutes a semi-supervised node-classification task (nodes corresponding to anything other than the complex molecule itself are unlabelled).
>
> Each molecule can be viewed as a separate graph, or the RDF can be viewed as a single graph comprising of a description of all molecules. If we train on batches of separate graphs and can fit all of the molecule graphs in memory, then these two are the same from the point of view of the training objective.
>
> [1] https://github.com/SmartDataAnalytics/DL-Learner/tree/develop/examples/mutagenesis
> [2] https://www.semanticscholar.org/paper/RDF2Vec%3A-RDF-Graph-Embeddings-for-Data-Mining-Ristoski-Paulheim/844d502387a3996f167b04e2e83117c30c22e752

---

### Public Comment · (anonymous) · 2018-09-29
**Interesting work!**

Nicely presented work with some interesting results!

Have you attempted to use alternative attentional mechanisms, e.g. the Transformer attention used by Vaswani et al. (NIPS 2017)? It has been found to give better performance on some tasks (see e.g. the Gated Attention Networks (GaAN) work of Zhang et al. (UAI 2018)).

I would recommend citing both the GaAN paper and the recent EGAT work of Gong & Cheng (2018):

https://arxiv.org/abs/1809.02709

as related works on deploying attention mechanisms on graphs. Especially, the EGAT also aims to incorporate edge-specific information (albeit, they consider arbitrary edge features, rather than distinct relational types).

---

> ### Author Response · Authors · 2018-10-01
> **Thank you for interest in our paper and the suggested articles.**
>
> Thank you for interest in our paper and the suggested articles.
>
> > Have you attempted to use alternative attentional mechanisms, e.g. the Transformer attention used by Vaswani et al. (NIPS 2017)? It has been found to give better performance on some tasks (see e.g. the Gated Attention Networks (GaAN) work of Zhang et al. (UAI 2018)).
>
> So far, we have only tried the style of attention used in the original GAT paper, and extending that style (using different softmaxes) to relational structures. We haven't tried to use the dot-product style of Vaswani et al., although this would be an interesting direction to investigate.
>
> The results of Zhang et al. [GaAN] show improvements on inductive tasks vs GAT, and so we would expect that extending GaAN to incorporate relational information (i.e. RGaAN) would yield similar gains on relational inductive tasks over RGAT on relational inductive tasks (i.e. Tox21 and others).
>
> On the RDF datasets, it is possible that RGaAN would perform better on AIFB than RGAT (and RGCN) - this would need to be due to the model having a better inductive prior for the task, although it is not immediately clear to us whether this is true. With MUTAG it is unlikely that any gains could be made using RGaAN vs RGCN for the same reasons that RGAT doesn't perform particularly well vs RGCN - as discussed in our results section. This is purely speculative and would need empirical validation.
>
> We will include a short discussion of alternative attention mechanisms in our related work section in an updated version of the manuscript.
>
> > I would recommend citing both the GaAN paper and the recent EGAT work of Gong & Cheng (2018): https://arxiv.org/abs/1809.02709 as related works on deploying attention mechanisms on graphs. Especially, the EGAT also aims to incorporate edge-specific information (albeit, they consider arbitrary edge features, rather than distinct relational types).
>
> Thank you for bringing this paper to our attention. We will include it in the related work section in an updated version of the manuscript as an alternative way of incorporating edge information into GAT models.
>
> Many thanks,
> The authors.

---

### Public Comment · ~Marc_Brockschmidt1 · 2018-10-01
**Relation to GGNN**

Hey,

This looks like interesting work. I was wondering if you had compared this to the Gated Graph Neural Network model we proposed a few years back (https://arxiv.org/abs/1511.05493). The model always supported different edge/relationship types, so it would be interesting to see how it fares in comparison. We've released a TF reference implementation of this (https://github.com/Microsoft/gated-graph-neural-network-samples), so it should be fairly doable to do a comparison.

On a related note, that implementation also has a 'use_propagation_attention' hyperparameter (in the _sparse.py model) that implements an attention mechanism over the incoming messages, i.e. the update rule is something like this:
  h'_v = GRU(h_v, \sum{u has k-edge to v} a_{u, k, v} m_{u, k, v})
  m_{u, k, v} = W_k * h_u
  a_{u,k,v} \propto exp((h_v * h_u^T) * a_k)
with W_k and a_k relationship-specific weights. This is not quite the same as the attention mechanism that you are proposing, so the experiences may differ for good reasons, and I implemented this quickly after seeing the GAT paper to see if the ideas would interact well with each other. However, in my version, I found it to be slow (due to the required softmax over dynamically-sized neighborhoods) and I never found it produce better results than the basic multi-layer GGNN model. I was wondering if you could share experiences with the implementation, and how you got the attention mechanism to work efficiently?

---

> ### Author Response · Authors · 2018-10-02
> **Details of implementation - hope this helps!**
>
> Thank you for your interest in our paper and comments.
>
> Hi Marc, thank you for your interest in our paper and comments.
>
> > I was wondering if you had compared this to the Gated Graph Neural Network model we proposed a few years back (https://arxiv.org/abs/1511.05493) ...
>
> We have not compared RGAT or RGCN to GGNNs, although it would be interesting to do so.
> In hindsight, recurrent approaches should have been included in our related work as an alternative way of incorporating relational information into a neural model.
> We will make this amendment in an updated version of the manuscript.
>
> In terms of performing the comparison to GGNNs, although we believe it is interesting and worthwhile, to perform a fair comparison, we would want to perform a full hyperparamter search for the model, and feel it lies slightly outside of the current scope of this work.
> That is, we set out to include relational properties into the GAT architecture of Veličković et al. https://arxiv.org/abs/1710.10903 and report our findings, comparing against existing baselines, in particular, its spectral counterpart - RGCN.
>
> > On a related note, that implementation also has a 'use_propagation_attention' hyperparameter (in the _sparse.py model) that implements an attention mechanism over the incoming messages, i.e. the update rule is something like this: ... I was wondering if you could share experiences with the implementation, and how you got the attention mechanism to work efficiently?
>
> Thank you for this - we refactored the code multiple times to achieve the performance required to run the experiments we presented.
> The speed of our layer is now essentially the same as our implementation of RGCN - up to a sparse softmax (which accounts for around half of the compute time of the forward and backward passes).
>
> We will be making our TensorFlow implementation of the model available, as well as the experimental setup to reproduce the Tox21 results.
> We may also implement this in PyTorch, although this is currently in-progress.
>
> The main optimisations we followed for the final implementation were: vectorise everything, compute everything only once, keep everything sparse if possible.
> This way, we end up in an implementation that is linear in the number of edges (across all relation types), and is fully vectorised.
> For comparison, an earlier implementation that did not have these optimisations suffered from a large amount of waiting time between the execution of ops, with end-to-end training time being anything from 20x-50x slower than the vectorised one.
>
> Schematically:
> - The nodes start in a dense representation (N,F)
> - The relational adj matrix is a SparseTensor with dense shape (N,RN)
> - The nodes get mapped by a single kernel (F,RHF') to (N,RHF'); this accounts for multiple heads H and multiple relation types R in a single dense (or basis decomposition) layer
> - We map the node features in (N,RHF') to 2x(N,RN,H) using the attention kernel (RH,F',2), and provide the feature base for the logits of the attention mechanism
> - Using the sparse adjacency matrix and tf.gather we can build up the features (E,H) for all incoming connections, and (E,H) for outgoing connections, where E is the number of edges across all relations
> - These are then added together with the LeakyRelu non-linearity applied, forming the logits of our eq (3).
> - Since we have not changed the order of any of the values, these can be placed back into a SparseTensor whose indices are those of the relational adj matrix (above).
> - For WIRGAT, this logits SparseTensor can be transformed into a shape (H,N,R,N), and a SparseSoftmax taken over the final dim, producing an attention SparseTensor - eq (4)
> - For ARGAT, this logits SparseTensor can be transformed into a shape (H,N,RN), and a SparseSoftmax taken over the final dim, producing an attention SparseTensor - eq (5)
> - The remaining step is to transform the attention SparseTensor into (H,N,RN) and sparse matrix multiply it with the transformed node representation (H,RN,F'), for each head - eq(6), eq(7)
> - We then concat or mean across relations, add bias and apply activation - eq (6), eq(7)
>
> In order to perform batched computation, we concatenate B sets of node features along their 0th dimension and produce a block diagonal sparse matrix containing B adjacency matrices.
> Unfortunately, to ensure that the indices are in the correct order for the forward pass, this concatenatenation process is expensive due to a sparse_reorder.
> If we were to optimise this further, we would start here.
>
> Hope this helps - we think that the implementation is one of the core contributions of our work and are looking forward to releasing it.
>
> The authors

---

### Public Comment · ~Michael_Bronstein1 · 2018-10-07
**generalizations of graph attention**

Interesting paper! I wonder how your approach compares to our recent extension of the graph attention mechanism in

Dual-Primal Graph Convolutional Networks, arXiv:1806.00770

---

> ### Author Response · Authors · 2018-10-08
> **Relation to DPGCNNs**
>
> Hi Michael, thank you for your interest in our work.
>
> We also find the approach of Dual-Primal Graph Convolutional Networks interesting.
> This is due to their property of allowing attention coefficients to be calculated on the basis of being common information propagators to a given node or from another given node - this treatment of an edge and the role it plays (in tandem with edges playing similar roles) isn't a treatment that is immediately accessible to the attention mechanisms of GaT or GaAN.
>
> In terms of how your approach compares with ours, we view DPGCNN as an improvement operator to GCN, providing an alternative way of calculating the logits to GAT for a GAT-style aggregation (in the same way that GaAN choses a dot-product key/value approach to compute the logits). The key difference with your setup is that the logits themselves are explicitly tied to edge feature representations constructed in the dual graph, giving the benefits we mention above.
>
> Our work on the other hand focuses on extending an attention mechanism that provides logits to incorporate the relation type specified by data source. In our study we form the logits in a relational generalisation of the additive-style of GAT, however, it would be equally valid (and interesting) to construct these logits from a relational extension of the dot-product style of GaAN, or - in the case of a relational DPGCNN - from the the edge features of an RGAT applied to a relational dual graph.
>
> We will include a note that a relational extension of DPGCNN should be investigated to compute the relational logits in an updated version of the manuscript.
>
> Many thanks,
> The authors.

---

### Author Response · Authors · 2018-11-26
**Summary of modifications**

Here is a summary of the modifications we have made to the paper:
- We have performed additional experiments using multiplicative attention. On the RDF tasks, we find that ARGAT is best paired with multiplicative attention, whereas WIRGAT is best paired with additive attention. On the Tox21 task we observe a much more significant difference coming from multiplicate attention when compared with its constant attention mode.
- We have evaluated RGCN under the same setup as RGAT. We have found that this narrows the gap between RGCN and RGAT, causing us to modify our conclusions and reframe the paper slightly - it is now presented as “an investigation into relational attention mechanisms”, rather than the presentation of a specific model we are advocating.
- The discussion in the introduction is more graph focussed.
- CDFs for the model are presented in appendix D, to aid insight into characteristic model behaviour, enabling future investigation. We provide some commentary around these results.
- Hypothesis tests are performed in appendix E, to aid discussion of whether any results in terms of model performance are significant. These tests are referred to in the new results section, which now compares additive and multiplicative attention against the retrained RGCN, rather than the performance of existing RGCN benchmarks.

Many thanks,
The authors.

---

### Meta-Review · Area_Chair1 · 2018-12-13

**Confidence:** 5
**Recommendation:** Reject

**Metareview:**

The authors propose an architecture for learning and predicting graphs with relations between nodes. The approach is a combination of recent research efforts into Graph Attention Networks and Relational Graph Convolutional Networks. The authors are commended for their clear and direct writing and presentation and their honest claims and their empirical setup. However, the paper simply doesn't have much to offer to the community, since the algorithmic contributions are marginal and the results unimpressive. While the authors justify the submission in terms of the difficult implementation and the extensive experiments, this is not enough to support its publication at a top conference. Rather, this could be a technical report.